# The Dynamic Role of Curcumin in Mitigating Human Illnesses: Recent Advances in Therapeutic Applications

**DOI:** 10.3390/ph17121674

**Published:** 2024-12-11

**Authors:** Md Shamshir Alam, Md Jamir Anwar, Manish Kumar Maity, Faizul Azam, Mariusz Jaremko, Abdul-Hamid Emwas

**Affiliations:** 1Department of Pharmacy Practice, College of Pharmacy, National University of Science and Technology, P.O. Box 620, Bosher, Muscat 130, Oman; 2Department of Pharmacology and Toxicology, College of Pharmacy, Qassim University, Buraydah 51452, Saudi Arabia; 3Department of Pharmacy Practice, MM College of Pharmacy, Maharishi Markandeshwar (Deemed to be University), Mullana, Ambala 133207, Haryana, India; 4Department of Medicinal Chemistry and Pharmacognosy, College of Pharmacy, Qassim University, Buraydah 51452, Saudi Arabia; 5Smart-Health Initiative (SHI) and Red Sea Research Center (RSRC), Division of Biological and Environmental Sciences and Engineering (BESE), King Abdullah University of Science and Technology (KAUST), Thuwal 23955, Saudi Arabia; 6Core Labs, King Abdullah University of Science and Technology (KAUST), Thuwal 23955, Saudi Arabia

**Keywords:** curcumin, turmeric, anti-inflammatory, anticancer, antibacterial, antifungal, antioxidant

## Abstract

Herbal medicine, particularly in developing regions, remains highly popular due to its cost-effectiveness, accessibility, and minimal risk of adverse effects. *Curcuma longa* L., commonly known as turmeric, exemplifies such herbal remedies with its extensive history of culinary and medicinal applications across Asia for thousands of years. Traditionally utilized as a dye, flavoring, and in cultural rituals, turmeric has also been employed to treat a spectrum of medical conditions, including inflammatory, bacterial, and fungal infections, jaundice, tumors, and ulcers. Building on this longstanding use, contemporary biochemical and clinical research has identified curcumin—the primary active compound in turmeric—as possessing significant therapeutic potential. This review hypothesizes that curcumin’s antioxidant properties are pivotal in preventing and treating chronic inflammatory diseases, which are often precursors to more severe conditions, such as cancer, and neurological disorders, like Parkinson’s and Alzheimer’s disease. Additionally, while curcumin demonstrates a favorable safety profile, its anticoagulant effects warrant cautious application. This article synthesizes recent studies to elucidate the molecular mechanisms underlying curcumin’s actions and evaluates its therapeutic efficacy in various human illnesses, including cancer, inflammatory bowel disease, osteoarthritis, atherosclerosis, peptic ulcers, COVID-19, psoriasis, vitiligo, and depression. By integrating diverse research findings, this review aims to provide a comprehensive perspective on curcumin’s role in modern medicine and its potential as a multifaceted therapeutic agent.

## 1. Introduction

The utilization of plants as medicinal remedies has been a cornerstone of human health across diverse societies and civilizations throughout history [1,2]. *Curcuma longa* (*C. longa*), commonly known as turmeric, belongs to the *Zingiberaceae* family and is widely recognized for its diverse applications worldwide. Native to tropical South Asia, turmeric has been traditionally used as a dye, a cleansing agent (e.g., before marriage), a food additive (spice), and a medicine [3].

The structure of curcumin, supported by its synthesis, was published as early as 1910 by Miłobędzka and coworkers [4]. Later, Lampe [5] introduced another synthetic method in 1918. This multi-step process began with ethyl acetoacetate and carbomethoxyferuloyl as the primary reagents. Later, Pabon [6] developed a more efficient and straightforward synthesis with improved yields (Figure 1). His method involved reacting acetyl acetone with boron trioxide (B_2_O_3_), substituted aromatic aldehydes, primary amine, and alkyl borate. With minor modifications, Pabon’s synthetic approach has been widely adopted by researchers worldwide, forming the basis for most curcumin syntheses [7,8,9,10].

Modern studies have confirmed that turmeric is very rich in antioxidants, which is in line with its historically known anti-inflammatory properties. Curcumin, the primary active compound in turmeric, along with other curcuminoids, such as dihydrocurcumin, tetrahydrocurcumin, bis-demethoxycurcumin, and demethoxycurcumin (Figure 1), has attracted considerable attention for its therapeutic potential [11,12,13,14]. These compounds have been extensively studied for their efficacy in preventing and treating a wide array of illnesses, including inflammatory diseases, cancer, diabetes, bacterial and fungal infections, ulcers, and liver and circulatory disorders, and as an anticoagulant agent (Figure 2) [11,12,13,14]. In addition, curcumin has been tested in clinical studies both alone and in conjunction with several other medications, including sulfasalazine, quercetin, piperine, lactoferrin, docetaxel, prednisone, gemcitabine, and pantoprazole [15].

Despite the promising therapeutic benefits demonstrated in both preclinical and clinical studies, the exact molecular mechanisms through which curcumin exerts its effects remain only partially understood. Curcumin’s ability to modulate multiple cellular signaling pathways suggests a multifaceted mechanism of action, which may contribute to its effectiveness as both an adjuvant and alternative therapy [14,16]. For instance, clinical trials have shown that curcumin can decrease inflammatory markers and enhance antioxidant capacities in healthy individuals [17], improve circulation [18], chelate metals [19,20], and support neurogenesis [21], among other benefits.

This review hypothesizes that the diverse pharmacological characteristics of curcumin, especially its antioxidant and anti-inflammatory properties, are critical in influencing the fundamental disease mechanisms involved in a variety of human illnesses. By targeting multiple cellular pathways, curcumin has the potential to improve therapeutic outcomes when used alone or in combination with other therapies.

To explore this hypothesis, the following sections will provide an in-depth analysis of recent studies on curcumin, focusing on its molecular mechanisms of action and therapeutic applications in various human diseases, including cancer, inflammatory bowel disease, osteoarthritis, atherosclerosis, peptic ulcers, COVID-19, psoriasis, vitiligo, and depression. Additionally, the review will examine synergistic responses and the implications of curcumin in diverse pathogenesis by modulating a variety of biomolecules and mediators.

## 2. Preclinical and Clinical Evidence for Therapeutic Applications of Curcumin

### 2.1. Cancer

Curcumin demonstrates significant anticancer properties by modulating key molecular signaling pathways, leading to apoptosis and inhibiting tumor growth and metastasis [22]. It effectively suppresses NF-κB by inhibiting IkBs, thereby reducing the expression of pro-inflammatory genes, like TNF-α (Figure 3) [23,24]. Additionally, curcumin downregulates AP-1, which is associated with anti-apoptotic proteins, promoting cancer cell death [25]. By targeting the IL-6/STAT3 pathway, curcumin disrupts oncogenic signaling, further inhibiting tumorigenesis [26,27]. These mechanisms underscore curcumin’s dual role as an anti-inflammatory and pro-apoptotic agent, enhancing its potential as both a standalone and adjunctive therapy across various cancer types, including pancreatic, prostate, breast, oral, lung, colorectal cancers, and multiple myeloma [28,29,30].

In colorectal cancer, curcumin addresses the complex etiology involving genetic, environmental, and inflammatory factors [31,32]. Clinical trials have shown that standardized curcuma extracts achieve pharmacologically active levels in colorectal tissues, reducing polyp size and number in familial adenomatous polyposis patients and preventing precancerous lesions in smokers without significant toxicity [15,33,34,35]. For instance, a dose-escalation study administering up to 3.6 g/day of curcumin demonstrated effective tissue concentrations and safety, while combinations with quercetin further enhanced therapeutic outcomes [19,33,35]. These findings highlight curcumin’s efficacy and safety, emphasizing the need for larger, randomized clinical trials to optimize its bioavailability and fully validate its therapeutic benefits in colorectal cancer treatment.

### 2.2. Inflammatory Bowel Disease

Curcumin demonstrates substantial anti-inflammatory effects by interacting with toll-like receptors and regulating mediators, such as MAPK and NF-κB, making it a promising therapeutic agent for inflammatory bowel disease (IBD) [36,37]. Clinical trials have shown that curcumin effectively induces and maintains remission in IBD patients. For example, studies report significant reductions in the inflammatory markers ESR and CRP in patients with ulcerative proctitis and Crohn’s disease following curcumin administration [38]. Additionally, maintenance therapy combining curcumin with sulfasalazine or mesalamine results in lower recurrence rates compared to a placebo [39], and case reports indicate that curcumin alongside prednisone can achieve clinical and endoscopic remission [40]. Another clinical trial demonstrated that curcumin supplementation at 1500 mg per day for eight weeks significantly improves clinical conditions and reduces inflammatory markers in individuals with mild-to-moderate ulcerative colitis [41]. Dose-dependent studies reveal that curcumin lowers levels of MAPK, NF-κB, and MMP3 while increasing IL-10 and reducing IL-1beta [42], and in vivo research supports its role in inhibiting the NLRP3 inflammasome [43]. However, curcumin’s inhibition of hepcidin may exacerbate anemia in IBD patients, necessitating careful monitoring [44,45]. These findings underscore curcumin’s potential as a complementary therapy for IBD, warranting further large-scale randomized trials to confirm its efficacy and safety.

### 2.3. Osteoarthritis

Osteoarthritis (OA) is a chronic inflammatory condition marked by joint degradation and cartilage damage, driven by dysregulated pro-inflammatory markers and cytokines, such as IL-1 and TNF-alpha. Curcumin emerges as a promising candidate due to its ability to modulate key inflammatory and catabolic pathways involved in OA pathogenesis. Preclinical studies demonstrate that curcumin reduces the expression of pro-inflammatory cytokines and matrix metalloproteinases, thereby protecting cartilage integrity and inhibiting disease progression [46,47,48,49,50]. Formulations, such as Advanced Ultrasol Curcumin and curcumin-loaded nanoparticles, have improved its bioavailability and therapeutic efficacy in animal models [51]. In addition, nanoparticles loaded with curcumin alleviated type II collagen loss, resulting in the reversal of elevated malondialdehyde levels in monosodium iodoacetate-induced knee OA in rodents [52]. Clinically, proprietary formulations, like Meriva™, have shown significant improvements in pain, joint function, and inflammatory markers in OA patients, while reducing the need for NSAIDs and minimizing adverse effects [53,54,55,56]. These outcomes suggest that curcumin not only addresses symptomatic relief but may also influence the underlying disease mechanisms. Integrating curcumin into OA management could offer a multifaceted approach by targeting inflammation, protecting cartilage, and enhancing patient quality of life [57]. Further large-scale randomized trials are warranted to fully establish its efficacy, optimize dosing regimens, and confirm its role as a viable adjunct therapy in osteoarthritis treatment protocols.

### 2.4. Atherosclerosis

Atherosclerosis is a chronic inflammatory condition of the vascular tissues and a major risk factor for cardiovascular diseases, primarily driven by oxidative stress that damages the vascular endothelium, triggering disease development [58]. Curcumin has emerged as a promising therapeutic agent due to its potent anti-inflammatory and antioxidant properties, which influence inflammatory cells and enzymes involved in atherosclerosis [59,60]. By scavenging reactive oxygen species (ROS) and enhancing the activity of endogenous antioxidant enzymes, such as glutathione peroxidase and superoxide dismutase, curcumin mitigates oxidative stress and protects vascular integrity [61,62,63,64,65]. Furthermore, curcumin modulates the TLR4/MAPK/NF-κB pathway, reducing the activity of pro-inflammatory M1 macrophages and lowering overall inflammation, thereby inhibiting the progression of atherosclerosis [66,67,68,69]. Clinical studies reinforce curcumin’s cardiovascular benefits. Randomized controlled trials have shown that curcumin supplementation improves lipid profiles by increasing high-density lipoprotein cholesterol (HDL-C) and decreasing plasma triglycerides (TGs) [70]. Additionally, curcumin (Theracurmin^®^) significantly decreased LDL-C levels in patients with chronic obstructive pulmonary disease (COPD), highlighting its role in reducing atherosclerotic risk factors [71]. However, some meta-analyses report no significant changes in total cholesterol or low-density lipoprotein cholesterol (LDL-C), indicating the need for more comprehensive research [72]. These findings underscore curcumin’s potential as a multifaceted adjunct therapy for atherosclerosis by targeting oxidative stress, inflammation, and lipid metabolism. Nonetheless, further large-scale randomized trials are essential to validate its efficacy and optimize dosing strategies for cardiovascular disease management.

### 2.5. Peptic Ulcer

Peptic ulcer disease arises from the disruption of the gastrointestinal (GI) tract protective barrier, primarily due to increased gastric acid secretion or decreased duodenal bicarbonate production. This disruption leads to mucosal injury and inflammation. Curcumin offers a promising therapeutic approach by reinforcing the mucosal barrier through its anti-apoptotic and antioxidant properties. Inhibition of caspase-3 by curcumin reduces gastric acid secretion and mitigates oxidative stress, thereby protecting the gastric mucosa [73]. Curcumin also exhibits antisecretory activity against nonsteroidal anti-inflammatory drugs (NSAIDs), enhancing gastroprotective effects [74]. Studies demonstrate that curcumin and bisdemethoxycurcumin decrease inducible nitric oxide synthase (iNOS) expression without affecting tumor necrosis factor-alpha (TNF-α), thereby reducing inflammation and acid secretion [75]. In rodent models, curcumin administration increases gastric juice pH, reduces neutrophil activity, and lowers oxidative stress markers, showcasing anti-ulcer efficacy [76]. Several formulations, such as chitosan–curcumin mixtures and omeprazole–curcumin-loaded hydrogel beads, further improve curcumin’s therapeutic outcomes by enhancing its antioxidant and anti-inflammatory activities [77,78,79]. Additionally, curcumin has demonstrated anti-Helicobacter pylori effects in mouse models, evidenced by significant reductions in serum IL-4, IFN-γ, somatostatin, gastrin, lipid peroxide, myeloperoxidase, and bacterial counts, alongside increased anti-H. pylori antibodies [80]. Clinically, turmeric supplementation has shown significant ulcer remission and symptom improvement, although antacids remain more effective in some cases [81,82]. The potential of curcumin as a complementary therapy for peptic ulcer disease is emphasized by these findings, which demonstrate that it can protect and repair the gastric mucosa through multifaceted mechanisms. However, larger randomized trials are necessary to fully validate its efficacy and optimize dosing regimens for clinical application.

### 2.6. COVID-19

COVID-19 is characterized by a hyperinflammatory state marked by elevated levels of colony-stimulating factors (G-CSF and GM-CSF), inflammatory markers (IL-1β, IL-6, IL-8, and TNF-α), and activated cytokine-tracing agents [83]. Central to its pathogenesis is the activation of the NOD-like receptor pyrin domain-containing 3 (NLRP3) inflammasome, which contributes to the catastrophic cytokine storm leading to acute respiratory distress syndrome (ARDS) and increased mortality, especially in patients with comorbidities, like diabetes and obesity [84,85,86,87]. Curcumin presents a multifaceted therapeutic potential by modulating these inflammatory pathways. It inhibits the NLRP3 inflammasome, thereby reducing the excessive inflammatory response [88]. Additionally, curcumin regulates Treg and Th17 cell populations, lowering inflammatory markers and mitigating immune dysregulation in COVID-19 patients [89].

Preclinical studies demonstrate curcumin’s ability to inhibit SARS-CoV-2 entry into cells, impede viral replication, and reduce inflammatory mediator production, which collectively diminish disease severity [90]. Clinical trials further support these findings, showing that curcumin supplementation improves oxygen levels, reduces hospital stays, and accelerates recovery in COVID-19 patients [91,92]. Moreover, combination therapies involving curcumin and other agents, such as piperine, have been shown to enhance clinical outcomes and reduce mortality rates [92]. Recent research also highlights specific curcuminoids, like Me08 and Me23, for their antiviral and anti-inflammatory effects, suggesting their potential in targeting both the virus and the host’s inflammatory response [93].

These integrated actions of curcumin underscore its potential as an effective adjuvant therapy for COVID-19, addressing both viral replication and the detrimental hyperinflammatory response. However, further large-scale randomized clinical trials are essential to validate curcumin’s efficacy, optimize dosing regimens, and fully integrate it into COVID-19 treatment protocols.

### 2.7. Psoriasis

Psoriasis is a chronic inflammatory skin condition characterized by scaly, red, and thick plaques that can affect any part of the body. Current treatments, such as UVB therapy and systemic medications, like methotrexate and cyclosporine, are often time-consuming and may pose risks to vital organs. Curcumin, renowned for its anti-inflammatory, antioxidant, and antimicrobial properties, emerges as an ideal herbal compound for managing psoriasis [94]. Curcumin exerts its therapeutic effects by modulating key inflammatory pathways, including the inhibition of NF-κB and the reduction of pro-inflammatory cytokines, like TNF-alpha, IL-17, and IL-22, thereby alleviating the inflammatory environment associated with psoriasis [95,96,97,98]. Additionally, curcumin inhibits the Kv1.3 potassium channel, which is crucial for T cell activation and proliferation, further mitigating psoriatic inflammation [99,100,101]. Moreover, the Lineweaver–Burk plot analysis demonstrated that curcumin is a non-competitive inhibitor of PhK with an inhibition constant (Ki) of 0.075 mM [102]. Advanced formulations, such as curcumin-loaded nanoparticles and SmartPearls, enhance skin penetration and therapeutic efficacy, resulting in improved anti-keratinization and a reduction in inflammatory markers [103,104,105,106]. Clinical studies demonstrate that curcumin supplementation significantly reduces psoriasis severity scores and inflammatory markers, though some trials indicate the necessity for larger, placebo-controlled studies to confirm these benefits [107,108]. These findings highlight curcumin’s potential as a multifaceted adjunct therapy for psoriasis, targeting both symptom relief and underlying inflammatory processes, warranting further comprehensive clinical research.

### 2.8. Vitiligo

Vitiligo is a chronic inflammatory skin condition characterized by the loss of melanocytes, resulting in white patches on the skin. Oxidative stress plays a significant role in melanocyte destruction, contributing to the disease’s etiology [109,110]. Curcumin, with its potent antioxidant and anti-inflammatory properties, emerges as a promising therapeutic agent for vitiligo. By reducing oxidative stress and modulating inflammatory pathways, curcumin helps protect melanocytes and promote skin repigmentation [111,112].

Preclinical research highlights curcumin’s ability to decrease advanced glycation end-products and activate the NRF2 antioxidant pathway, enhancing cellular protection against oxidative damage [111,112]. These findings underscore curcumin’s potential as a complementary therapy for vitiligo, addressing both oxidative stress and inflammation to restore skin pigmentation. Further large-scale randomized trials are essential to validate curcumin’s efficacy and optimize treatment protocols for vitiligo patients.

Clinical studies support curcumin’s efficacy in managing vitiligo. In a trial with ten patients, the combination of tetrahydrocurcuminoid cream and narrowband UVB (NB-UVB) therapy resulted in significant repigmentation compared to baseline, with enhanced outcomes observed in the combination group [109]. Additionally, turmeric cream applied twice daily for four months significantly reduced lesion size and improved vitiligo scores compared to placebo controls [113]. A case study further illustrated spontaneous repigmentation following the consumption of a curcumin-based herbal remedy [114].

### 2.9. Depression

Depression is increasingly recognized as being closely linked to inflammation, where dysregulated innate and adaptive immune responses contribute to its onset [115,116]. Curcumin, renowned for its anti-inflammatory and antioxidant properties, presents a promising therapeutic option for alleviating depressive symptoms. By inhibiting the NLRP3 inflammasome, curcumin reduces the expression of pro-inflammatory cytokines, thereby mitigating the inflammatory processes associated with depression [117]. Preclinical studies demonstrate that curcumin administration reverses depressive-like behaviors in stressed rodents by restoring neurotransmitter levels, including serotonin, norepinephrine, and dopamine, and modulating the serotonergic system through 5-HT receptors [118,119,120,121,122]. Additionally, curcumin enhances the brain-derived neurotrophic factor (BDNF) pathway, promoting neuronal survival and synaptic plasticity, which are critical for mood regulation [123,124].

Enhanced formulations, such as curcumin-coated nanoparticles, have further improved its bioavailability and therapeutic efficacy, resulting in significant behavioral and biochemical improvements in animal models of depression [125,126,127]. Clinical trials support these findings, with curcumin supplementation showing reductions in inflammatory markers and improvements in depressive symptoms [128,129,130]. However, some studies highlight the need for larger, randomized controlled trials to confirm curcumin’s antidepressant efficacy and optimize dosing strategies [129,130]. These integrated actions of curcumin underscore its potential as a multifaceted adjunct therapy for depression, targeting both inflammatory pathways and neurochemical imbalances to enhance mental health outcomes.

### 2.10. Diabetes Mellitus

Diabetes mellitus, particularly type 2 diabetes (T2DM), is a chronic metabolic disorder characterized by impaired glucose metabolism, insulin resistance, and associated complications, such as neuropathy, cardiovascular, and renal damage. Curcumin, with its multifaceted bioactive properties, has been extensively investigated for its potential to manage diabetes and its complications through mechanisms including improved lipid metabolism, reduced inflammation, antioxidant effects, and prevention of cell death [131,132,133].

Curcumin administration has demonstrated significant benefits in clinical trials. In a randomized, double-blind, placebo-controlled study, nano-curcumin (80 mg/day) for three months in T2DM patients resulted in notable reductions in glycosylated hemoglobin (HbA1c), fasting blood glucose (FBG), triglycerides (TG), and body mass index (BMI), along with improvements in low-density lipoprotein cholesterol (LDL-C) levels [134]. Additionally, curcumin administration at 300 mg/kg for three months in diabetic obese patients significantly reduced fasting blood sugar, insulin resistance, HbA1c, serum triglycerides, and free fatty acids (FFAs), while increasing lipoprotein lipase activity, highlighting its efficacy in lipid metabolism modulation [135]. Proprietary formulations, like Theracurmin^®^, have shown efficacy in reducing α1-antitrypsin-LDL levels and increasing adiponectin in non-insulin-dependent diabetic patients [136]. Additionally, combining curcumin with piperine significantly improved glycemic control and hepatic parameters, although effects on high-sensitivity C-reactive protein (hs-CRP) were inconclusive due to study limitations [137].

Curcumin also plays a role in mitigating diabetic complications. Topical application of curcumin ointments significantly reduced the size of diabetic foot ulcers, enhancing wound healing through sustained local concentrations and anti-inflammatory effects [138,139]. These findings underscore curcumin’s potential as a comprehensive adjunct therapy for managing diabetes and its complications, although further large-scale, long-term randomized trials are necessary to fully establish its efficacy and optimize dosing strategies.

### 2.11. Malaria

Malaria, caused by *Plasmodium* species, remains a significant global health challenge, particularly due to the emergence of drug-resistant strains. Curcumin, a polyphenolic compound from turmeric, has shown promising antimalarial potential through multiple mechanisms. Preclinical studies indicate that curcumin inhibits the histone acetyl transferase (HAT) enzyme and generates reactive oxygen species (ROS) within the malaria parasite, leading to cellular damage and reduced parasitemia [140,141]. Notably, curcumin effectively targets chloroquine-resistant *Plasmodium falciparum* with an IC₅₀ of approximately 5 µM and significantly decreases parasitic loads and increases survival rates in *Plasmodium berghei*-infected mice [142]. Additionally, curcumin enhances the efficacy of traditional antimalarial drugs, like α,β-arteether, resulting in 100% survival and protection against recrudescence when combined [142,143].

Mechanistically, curcumin mitigates cerebral malaria by inhibiting pro-inflammatory cytokines, such as TNF-α, IL-1β, IL-6, and IFN-γ, thereby preventing blood–brain barrier disruption and neuronal injury [144,145]. Several curcumin formulations, including PLGA nanoparticles [146], nano-emulsions [147], hydrogel nanoparticles [148], solid–lipid dispersions [149], chitosan nanoparticles [150], polymeric capsules [151], and lipoidal drug delivery systems [152], have demonstrated improved bioavailability and greater efficacy in reducing inflammatory markers, preventing infected red blood cell sequestration in the brain, and exhibiting antimalarial activity in experimental animals. Furthermore, combinations of curcumin with bioavailability enhancers, like piperine, have shown enhanced antimalarial effects by increasing ROS production, which is cytotoxic to *Plasmodium* parasites [153].

Despite the promising preclinical outcomes, clinical studies are limited. Future research should focus on comprehensive clinical trials to validate curcumin’s antimalarial efficacy, optimize dosing regimens, and explore its potential to counteract drug resistance, thereby contributing to sustained malaria control and eradication efforts.

### 2.12. Antimicrobial Properties

Curcumin, the primary curcuminoid in turmeric, has long been recognized for its antimicrobial properties, complementing its well-documented anti-inflammatory and antioxidant effects [154,155]. Curcumin combats bacterial infections through multiple mechanisms, including the inhibition of FtsZ, a crucial enzyme for bacterial cell division, and the disruption of bacterial cell membranes due to its amphipathic nature [156,157]. These actions render curcumin effective against both Gram-positive and Gram-negative bacteria, such as *Escherichia coli* and *Staphylococcus aureus* [156,157]. Additionally, curcumin impedes biofilm formation in pathogens, like *Pseudomonas aeruginosa*, enhancing the efficacy of conventional antibiotics and offering a promising strategy against antibiotic-resistant strains [158,159].

Beyond antibacterial effects, curcumin exhibits antiviral activity, demonstrated by its ability to interfere with viral entry and replication processes in *Herpes Simplex Virus type 1* (HSV-1), thereby reducing viral load and infection severity [160]. Curcumin also shows potential against *Helicobacter pylori*, effectively lowering bacterial counts and mitigating infection-related inflammation in preclinical models [161].

However, the clinical application of curcumin is limited by its poor bioavailability, rapid metabolism, and systemic elimination. Advances in nanotechnology have led to the development of nanocurcumin formulations, which significantly enhance curcumin’s stability, absorption, and therapeutic efficacy [162]. These innovative delivery systems not only improve antimicrobial activity but also enable synergistic interactions with existing antimicrobial agents, reducing the required dosages and minimizing side effects [163,164].

Despite the promising preclinical outcomes, comprehensive clinical studies are necessary to fully establish curcumin’s efficacy and safety as an antimicrobial agent. Future research should focus on optimizing curcumin formulations and exploring its potential as an adjunct therapy to overcome the growing challenge of antimicrobial resistance.

### 2.13. Neuroprotective Properties

Neurodegenerative disorders (NDs), such as Alzheimer’s disease (AD) and Parkinson’s disease (PD), are characterized by the progressive loss of neurons and impaired neural function, significantly impacting the aging population [165]. Current pharmacological treatments for NDs are limited by their inability to halt disease progression and their adverse side effects, prompting the exploration of alternative therapeutic strategies. Curcumin, a bioactive compound from turmeric, has emerged as a promising neuroprotective agent due to its multifaceted mechanisms of action.

Curcumin mitigates neuronal degeneration by suppressing oxidative stress, inflammation, cell death, and mitochondrial dysfunction [165]. It activates the NRF2 pathway by dissociating from Keap1, leading to the increased expression of antioxidant enzymes, such as heme oxygenase-1 (HO1) and NAD(P)H–quinone oxidoreductase 1 (NQO1), while inhibiting pro-inflammatory cytokines, like TNF-alpha, IL-6, and IL-1beta [166]. This dual action enhances neuronal survival and plasticity, critical for combating neurodegeneration.

In animal models, curcumin has demonstrated significant neuroprotective effects. For instance, chronic administration of curcumin in stressed rats reduced IL-1beta expression and neuronal death in the ventromedial prefrontal cortex, correlating with decreased depressive behavior [167]. Additionally, curcumin enhanced neurotransmitter levels and modulated the serotonergic system, contributing to its antidepressant effects [168]. Nanotechnology-based formulations, such as curcumin-coated nanoparticles, have further improved its bioavailability and efficacy, leading to enhanced behavioral and biochemical outcomes in rodent models of neurodegeneration [169,170,171].

In PD models induced by 6-hydroxydopamine, curcumin protected dopaminergic neurons, improved memory function, and increased dopamine and acetylcholine levels in the substantia nigra, while reducing oxidative stress and upregulating neurotrophic factors, like BDNF and TrkB [172,173,174]. Similarly, in AD models, curcumin reduced beta-amyloid plaque formation, oxidative stress, and neuronal damage, demonstrating its potential to slow disease progression [175,176,177,178]. Furthermore, the transcriptional modification of dysfunctional microglial miR-155 reduced retinal inflammation and related vasculopathy by decreasing the generation of Clec7a/Galectin-3+ microglia, which led to neurodegenerative outcomes [179]. Interestingly, curcumin at a lower dose stimulated microglial migration to and phagocytosis of amyloid plaques, decreased miR-155-mediated neurodegenerative phenotype, and reduced amyloid stress in mouse brains [180]. Clinical studies in humans have shown that bioavailable curcumin formulations, such as Theracurmin^®^ and CurQfen^®^, improve cognitive function and reduce biomarkers of neuronal degeneration, although some studies report inconclusive results, highlighting the need for larger trials [181,182,183].

Overall, curcumin’s ability to enhance antioxidant defenses, suppress inflammatory pathways, and promote neuronal health underscores its potential as a neuroprotective agent. Continued research, particularly large-scale clinical trials, is essential to fully establish curcumin’s efficacy and optimize its therapeutic applications in neurodegenerative diseases.

## 3. Curcumin’s Bioavailability: Challenges and Innovative Formulations

Despite the numerous advantageous effects of curcumin, its oral bioavailability continues to be a significant constraint in curcumin therapy, necessitating further research to resolve this problem. Curcumin has been demonstrated to have an exceptionally poor bioavailability, as evidenced by numerous studies that have demonstrated either undetectable or extremely low concentrations in plasma and other tissues [184]. The primary factors contributing to low bioavailability are insufficient absorption, swift first-pass metabolism, limited tissue penetration, chemical instability, and rapid elimination [185].

To overcome these obstacles, various strategies have been employed to enhance curcumin’s bioavailability. These includes the formulations of curcumin as nanoparticles, lipid formulations, drug complexation, and the use of structural analogs, such as turmeric oil [186,187]. Combining curcumin with 20 mg of piperine significantly boosts its bioavailability, increasing absorption by 2000%, resulting in a raised plasma concentration within 1 h of co-administration in both rats and humans without producing adverse effects [188]. The stability of curcumin complexes with metals was improved, which contributed to the enhancement of biological activities, including antioxidant, anti-inflammatory, antimicrobial, antineoplastic, neuroprotective, and anti-ulcer activity, in comparison to unformulated curcumin [189].

Nanotechnology-based formulations have shown promise in increasing the bioavailability and therapeutic efficacy of curcumin. For example, N-carboxymethyl chitosan-coated solid lipid nanoparticles of curcumin exhibited a 6.3-fold increase in oral bioavailability and a 9.5-fold increase in lymphatic absorption when contrasted with curcumin solution [190]. These findings suggest that NCC-SLN could serve as an effective oral delivery system for curcumin, improving both absorption and distribution. Currently, a variety of formulations, including BioCurc, Cavacurcmin, CurcuWIN, Hydrocurc, Meriva, Nanocurcumin, Novasol, Theracurmin, and Turmipure Gold, are available in the global market with the potential to enhance bioavailability and solubility. Emulsifiers, such as polyethoxylated hydrogenated castor oil, carbohydrate complexes, lipoidal/phospholipid complexes, and polysorbates, are commonly used in these preparations to further improve bioavailability and solubility [191]. In addition, poloxamers offer an intriguing approach to enhance the steric stability and assimilation of conventional liposomes. Curcumin’s bioavailability was elevated to 43.3% in the pluronic-modified liposomal formulation, as compared to 26.9% with the standard liposomal formulation [192]. These formulations have been shown to significantly enhance the bioavailability of curcumin, making it more effective for therapeutic use in various chronic diseases.

## 4. Safety Profile of Curcumin

According to the results of safety investigations, curcumin is generally considered safe for human consumption. It has also been revealed by several studies that curcuminoids, including curcumin, and the extracts of Curcuma longa do not elevate the incidence of adverse events [193,194]. Curcumin is safe for human consumption, even at comparatively high doses, as support from studies conducted in humans indicate that healthy volunteers can tolerate doses of up to 8 g/d and even 12 g/d. Furthermore, a meta-analysis of 6 human studies including 172 participants found that curcuminoids substantially reduced circulating C-reactive protein levels when compared to placebo, with no adverse effects [195]. Studies on its potential toxicity indicated that it is generally safe, even at larger quantities (up to 12 g in human) [196,197]. According to toxicological studies, curcumin does not cause any discernible mutagenic, teratogenic, or subchronic toxicity damage [193,198]. Furthermore, research conducted by a study group stated that the administration of 8 g per day of curcumin plus gemcitabine to pancreatic cancer patients was found to be well-tolerated and safe [20,21]. However, curcumin and its derivative, bisdemethoxycurcumin, have been reported to prolong the aPTT and PT, and inhibited the activity of thrombin and factor Xa [199]. Therefore, curcumin should be used cautiously in patients with bleeding disorders and in pregnant and nursing women. Curcumin’s advantageous effects are evident; however, it is crucial to evaluate its antiplatelet properties, as it has the potential to enhance the efficacy of other antiplatelet medications, including ginkgo biloba, garlic, and other anticoagulants [200], which could result in a significant risk of hemorrhagic complications. Notably, the inappropriate bioavailability of the administered formulation could also be considered, as the various formulations achieved varying plasma concentrations, which may also affect these interactions. Despite its anti-inflammatory properties, it has been recognized as an allergen causing contact dermatitis in some individuals [201]. Consequently, it is imperative that patients and dermatologists remain vigilant regarding potential allergic reactions.

## 5. Probable Mechanisms of Action of Curcumin

Curcumin’s efficacy in preventing the development of disease is shown, e.g., by its ability to efficiently counteract the effects of reactive oxygen species (ROS) and nitrogen species [202]. Curcumin inhibits lipid peroxidation and peroxide-induced DNA damage via the inhibition of oxidative stress by reducing the production of ROS, including the generation of highly reactive free radicals, such as hydroxyl groups, univalent anion oxygen, and singlet oxygen, etc. [203]. Nevertheless, the inflammatory response can be induced by oxidative stress, which in turn generates a greater proportion of free radicals that can further exacerbate oxidative stress. Cardiovascular ailments, diabetes, and arthritic conditions are among the conditions that can result from this cycle, as chronic inflammation can be induced by oxidative stress [204]. Curcumin modulates many signaling molecules to exert powerful anti-inflammatory and anti-carcinogenic effects [205]. Numerous crucial elements in cellular signal transduction pathways connected to proliferation, differentiation, and malignant transformations have been demonstrated to be inhibited by curcumin (Table 1) [206].

Furthermore, a multitude of studies have reported the implications of increased prostaglandin biosynthesis following upregulated cyclooxygenase-2 (COX-2) expression, which leads to activation of c-Jun/AP-1 and protein kinases in inflammatory conditions [207]. However, reports suggest that curcumin’s anti-inflammatory properties contribute to its inhibitory effect on enzymes, like 5-lipooxygenase and COX-2 [208]. In addition, curcumin has been reported to suppress lipopolysaccharide-induced increased expression of inflammatory genes, such as COX-2 and iNOS, in RAW 264.7 macrophages cells [209]. Furthermore, curcumin alleviated the carrageenan-induced increase in inflammatory cytokines, such as IFN-γ, IL-1β, IL-6, IL-13, and IL-17, through suppressing the NF-κB/COX-2 pathway and inhibiting iNOS expression [210].

**Table 1 pharmaceuticals-17-01674-t001:** Summary of the mechanisms/actions of curcumin and the main findings from the studies.

S. No.	Pathway/Methods	Study Type	Main Findings	References
1.	DPPH scavenging method	In vitro	Curcumin and nanosuspension exhibited more potency in scavenging of superoxide free radicals followed by demethoxycurcumin and bisdemethoxycurcumin.	[211,212,213]
2.	Radiation-induced lipid peroxidation DFT studies	In vitro, in vivo, and in silico	Curcumin inhibited lipid peroxidation by 82% and dimethoxy curcumin by 24%. In curcumin, the hydrogen of -OH is more labile for separation than the hydrogen of -CH(2).	[214]
3.	ABTS and DMPD radical scavenging assay	In vitro	Curcumin exhibited free radical scavenging activity against DPPH, ABTS, DMPD, superoxide anion free radical, and H_2_O_2_, as well as for ferrous (Fe^2+^) ion chelation and ferric ion (Fe^3+^) reduction.	[62]
4.	Styrene oxidation assay	In vitro	Curcumin produced phenolic chain-breaking antioxidant activity.	[203]
5.	Phosphomolybdenum peroxidation and linoleic acid peroxidation assay	In vitro	Curcumin exhibited maximum antioxidant activity followed by demethoxycurcumin and then bisdemethoxycurcumin.	[215]
6.	Keap1/Nrf2/ARE signaling pathway	In vitro	Curcumin demonstrated resistance to oxidizing agents by activating the Nrf2-Keap1 pathway and boosting the activity of antioxidant enzymes.	[216]
		In vivo	Curcumin demonstrated Nrf2 activation by inhibiting the upregulation of Keap1 induced by inflammatory signals, thereby contributing to its antioxidant benefits and alleviating insulin resistance in high-fat-diet obese mice.	[217]
7.	PI3K/Akt-1/mTOR signaling pathway	In vitro	Curcumin stimulates autophagy by inhibiting the PI3K/AKT/mTOR signaling pathway, which is achieved by at least partially inhibiting the phosphorylation of both Akt and mTOR.	[218,219]
8.	Modulation of inflammatory cytokines	In vivo	A curcuminoid preparation (180 mg/day) produces a significant reduction in TNFα, IL-6, substance P, hs-CRP, CGRP, and TGF-β as compared to a placebo control.	[220,221]
		In vivo	In patients with metabolic syndrome, the administration of curcumin (1 g/day) once daily results in a substantial decrease in serum levels of inflammatory cytokines.	[222]
		In vitro	Curcumin inhibited the release of inflammatory cytokines, including IL-8, monocyte inflammatory protein-1 (MIP-1α), monocyte chemotactic protein-1 (MCP-1), interleukin-1β (IL-1β), and tumor necrosis factor-α (TNF-α), from lipopolysaccharide-induced monocytes and macrophages extracted from the peripheral blood and alveoli of humans, respectively.	[223]
		In vitro	Curcumin reduces LPS-induced NO and pro-inflammatory cytokine production in microglial cells.	[224]
		In vitro	Curcumin suppresses PHA-induced T-cell growth, production of IL-2, NO generation, and LPS-induced NF-κB, while increasing NK cell cytotoxicity.	[225]
9.	Modulation of growth factors	In vitro	Curcumin suppressed the EGF/EGFR signaling pathway, which in turn prevented hyperglycemia-driven EGF-induced pancreatic cancer from migrating and invading by downregulating the ERK and Akt signaling molecules.	[226]
		In vitro	Curcumin suppresses angiogenesis by inhibiting FGF-induced neovascularization in cultured corneal cells.	[227]
		In vivo and in vitro	Both in vitro and in vivo models exhibited the angiogenesis-inhibiting effects of curcumin through its inhibitory action on VEGF and VEGFR2.	[228,229]
10.	Modulation of growth factor receptors	In vitro	Curcumin induces autophagy in both AR-positive and AR-negative prostate cancer cells by reducing the activity and expression of the androgen receptor (AR) and related cofactors AP-1, NF-κB, and CBP.	[230,231]
		In vitro	The Her2-Akt signaling pathway is suppressed by curcumin (18 μmol/L), resulting in the inhibition of cell proliferation in the human breast cancer cell line. Furthermore, the low-dose (1.5 μmol/L) combination of curcumin and lapatinib resulted in a more potentiated inhibitory effect on the Her2-Akt signaling pathway and the inhibition of cancer cell growth.	[232]
		In vitro	Curcumin inhibits EGFR by repressing the expression of genes and proteins in the EGFR-PI3K-AKT pathway, thereby overcoming Lenvatinib’s resistance in hepatocellular carcinoma cells.	[233]
		In vivo and in vitro	Curcumin enhanced the degradation of EGFR protein, thereby suppressing its activation. This, in turn, resulted in the reversal of gefitinib in non-small cell lung cancer in both in vitro and in vivo conditions.	[234]
11.	Modulation of enzymes	In vitro and in vivo	Curcumin treatment produces inhibition of NO production, reduced expression of iNOS and COX-2 by inhibiting ERK 1/2, and p38 activation in RAW 264.7 macrophages. Additionally, curcumin produces reversal of CPA-induced changes in body weight, immunoglobulins, and NK cell activity in mice.	[235]
		In vitro and in vivo	Addition of curcumin (5–10 microM) to epidermal microsomes produces inhibition of arachidonic acid metabolism into PGE2, PGF2α, and PGD2. Topically applied curcumin results in inhibition of the activity of LOX and COX in epidermal inflammation.	[236]
		In vivo	Curcumin is less ulcerogenic than phenylbutazone and exerts anti-inflammatory activity that is comparable to that of phenylbutazone. It also impeded the increased levels of SGOT and SGPT that were induced by inflammation.	[237]
		In vitro	Curcumin inhibits the AMPK-MAPK (mitogen-activated protein kinase) and PKC pathways by reversing PMA-induced PKC activation and suppressing chronic AMPK activation, resulting in decreased synthesis of EMMPRIN, MMP-9, and MMP-13.	[238]
12.	Modulation of adhesion molecules	In vitro	Curcumin demonstrated anti-inflammatory activity by suppressing monocytic THP-1 adhesiveness by inhibiting the TNF-α induced ICAM-1 expression in the human keratinocyte cell line.	[239]
		In vivo	Curcumin inhibited the formation of glial scars by preventing the production of MIP1α, IL-2, and CCL5 and by reducing NF-κB activation.	[240]
		In vitro	Curcumin modulates the PI3K/Akt/MAPK/NF-B pathway to suppress the expression of VCAM-1 induced by tumour necrosis factor-α (TNFα)/lipopolysaccharide (LPS) in human intestinal microvascular endothelial cell culture.	[241]
13.	Modulation of apoptosis-related proteins	In vitro	Increased apoptosis and growth inhibition were observed in HT-29 cells that were treated with curcumin. This was accompanied by corresponding changes in apoptosis-related proteins, including a decrease in the expression of Bcl-2, Bcl-xL, and survivin, as well as an increase in Bax and Bad.	[242]
14.	Modulation of cell cycle proteins	In vitro	The expression levels of PCNA, cyclin D1, and Bcl-xL were reduced by curcumin, which prevented human keratinocyte cell differentiation by arresting them in the G1/S phase.	[243]
		In vitro	Curcumin nanoparticles triggers cellular arrest at the G1/S and G2/M phases, as well as cell death, by inhibiting several cell cycle proteins, including cyclin E in MDA-MB-231 cells.	[244]
15.	Modulation of transcription factors and their signaling pathways	In vivo	Curcumin demonstrated an antidepressant effect by inhibiting the activation of NF-κB and reducing the expression of pro-inflammatory cytokines.	[119]
		In vitro	Curcumin suppresses inflammation induced by LPS by controlling microglia polarization (M1/M2), balancing TREM2/TLR4, and inhibiting NF-κB activity.	[37]
		In vitro	Curcumin lowers cellular viability in human and rat glioma cell lines, which is associated with the suppression of the AP-1 and NFkappaB signaling pathways by preventing JNK and Akt activation.	[245]
16.	Ca2+/CaN/NFAT/IL-2 signaling pathway	In vitro	Curcumin reduced inflammation by inhibiting T cell-mediated Ca2+ mobilization and NFAT-regulated IL-2 and NF-κB production in freshly separated T cells and the Jurkat T leukemia cell line.	[246]
17.	CD95/CD95L apoptotic signaling pathway	In vitro	Curcumin induces apoptosis in colorectal cancer cells via altering the Fas-regulated extrinsic pathway, activating caspase 8 and TRAIL binding to death receptors (DR), and upregulating DR5 proteins.	[247]
18.	c-Jun N-terminal kinases 1/2 (JNK1/2) pathways	In vitro	Curcumin inhibits MEKK1-induced JNK activation via interfering with upstream signaling molecules, resulting in the inhibition of AP-1 and NF-kappaB signaling and significant anti-inflammatory and anticancer effects.	[248]
19.	FABP5/ PPARβ/δ pathway	In vitro	Curcumin suppresses the expression of FABP5 and PPARβ/δ, thereby overcoming the resistance to retinoic acid in MDA-MB-231 and MD-MB-468 cell lines (triple-negative breast cancer).	[249]
20.	Modulation of PPARγ/NF-κB signaling pathway	In vitro and in vivo	Curcumin prevents cigarette smoke extract (CSE)-induced inflammation both in vivo and in vitro, possibly by upregulating PPARγ and inhibiting NF-κB activation.	[250]
		In vitro and in vivo	In in vitro and in vivo models, curcumin suppresses PPAR activation, inhibits NF-κB p65 translocation, and improves the increased expression of MCP-1 and MUC5AC induced by OVA and IL-4.	[251]

**Abbreviations:** ABTS, 2,2′-azino-bis(3-ethylbenzothiazoline-6-sulfonic acid); Akt-1, alpha serine/threonine-protein kinase-1; AMPK, 5′ AMP-activated protein kinase; AP-1, activator protein 1; AR, androgen receptor; ARE, antioxidant response element; Bcl-2, B-cell lymphoma-2; CaN, calcineurin; CBP, CREB binding protein; CCL5, chemokine (C-C motif) ligand 5; CGRP, calcitonin gene-related peptide; COX-2, cyclooxy-genase-2; CPA, cyclophosphamide; DFT, discrete Fourier transform; DMPD, N,N-Dimethyl-p-phenylene diamine; DPPH, 2,2-Diphenyl-1-picrylhydrazyl; EGF/EGFR, epidermal growth factor (EGF) receptor (EGFR); EMMPRIN, extracellular matrix metalloproteinase inducer; ERK1/2, extracellular signal-regulated protein kinases 1 and 2; FABP5, fatty acid binding protein 5; FGF, fibroblast growth factor; Her2, human epidermal growth factor receptor 2; hs-CRP, high-sensitivity C-reactive protein; ICAM-1, intercellular adhesion molecule 1; IL-1/6/8, interleukin-1/6/8; iNOS, inducible nitric oxide synthase; Keap1, Kelch-like ECH-associated protein 1; LOX, lysyl oxidase; LPS, lipopolysaccharide; MAPK, mitogen-activated protein kinases; MCP-1, monocyte chemoattractant protein 1; MIP-1α, macrophage inflammatory protein-1 alpha; MMP-9/13, matrix metalloproteinase-9/13; mTOR, mammalian target of rapamycin; NFAT, nuclear factor of activated T-cells; NF-kB, nuclear factor kappa-light-chain-enhancer of activated B cells; NO, nitric oxide; Nrf2, nuclear factor-erythroid 2-related factor 2; PCNA, proliferating cell nuclear antigen; PHA, phytohaemagglutinin; PI3K, phosphatidylinositide 3-kinases; PKC, protein kinase C; PMA, phorbol 12-myristate 13-acetate; PPARβ/δ, peroxisome-proliferator-activated receptor β/δ; PPARγ, peroxisome-proliferator-activated receptor γ; SGOT, serum glutamic oxaloacetic transaminase; SGPT, serum glutamate pyruvate transaminase; TGF-β, transforming growth factor beta; THP-1, human leukemia monocytic cell line; TLR4, Toll-like receptor 4; TNFα, tumor necrosis factor alpha; TRAIL, tumor necrosis factor (TNF)-related apoptosis-inducing ligand; TREM2, triggering receptor expressed on myeloid cells 2; VEGF, vascular endothelial growth factor; VEGFR2, vascular endothelial growth factor receptor-2.

Figure 3, illustrates the action mechanisms of curcumin in diverse pathologic conditions. Nuclear factor-kappa B (NF-kB) is a transcription factor that regulates the expression of numerous genes, including pro-inflammatory genes. It is a critical mediator between inflammation and the development of a variety of human diseases. Its activity is strictly regulated by a variety of mechanisms (Figure 3) and it functions in the regulation, activation, and maintenance of innate white blood cells, such as T- and B lymphocytes [86]. Tumor necrosis factor alpha (TNF-α) is a pleiotropic cytokine that is a significant regulator of inflammatory processes [252]. It is implicated in the pathogenesis of a wide range of pathologic conditions, including neurologic [253], autoimmune [254], and endocrine disorders [255], as well as obesity [256]. As shown in Figure 3, curcumin inhibits the expression of inflammatory genes by direct inhibitory action on NF-κB [257] and also by inhibiting the TNFα signal transduction pathway [258]. In addition, curcumin inhibits TLR4 homodimerization required for activation of the MyD88 adaptor protein. This suppresses downstream signaling molecules, like IRAK1 and IRAK4 kinases, preventing TAK1-mediated activation of the IKK complex and activation of NF-κB [259]. The inhibition of NF-κB pathways exerts modulatory action on cellular oxidative stress by regulating the production of cytokines, adhesion molecules, COX-2, iNOS [260], HO-1, and NQO-1 [261], which is crucial for the adipocyte hyperplasia and hypertrophy leading to obesity. NAD(P)H–quinone oxidoreductase 1 (NQO1) is a key enzyme in the antioxidative system, catalyzing the reduction of two electrons, which takes place in different endogenous and exogenous quinones where flavin adenine nucleotides (FAD) acts as a cofactor. The promoter domain of the NQO1 gene encodes antioxidant response element (ARE) sequences, which have been shown to be regulated by nuclear factor (erythroid-derived 2)-like 2 (NRF2) [262]. Curcumin may activate the NRF2 pathway, which may reduce the expression of cytokines that promote inflammation, such as MCP-1, TNF-a, IL-6, and IL-1β, and induce antioxidant enzymes [263,264]. Furthermore, curcumin exerts direct inhibitory action on adipocyte hyperplasia and hypertrophy [265]. The vast majority of tumor therapies rely on the direct destruction of cancer cells or promoting the action of antitumor immune cells, such as natural killer (NK) cells and cytotoxic CD8+ T lymphocytes (CTLs) [266]. Interestingly, curcumin exerts direct antitumor and cytotoxic action by elevating NK cells [267], cytotoxic CD8+ T cells [268], and INFγ [269], and by reducing the activity of Tregs and MDSC in tumor cells [270]. Also, curcumin suppresses tumor progression by reducing Tregs and MDSC [271]. It directly suppresses the phosphorylation of AKT, leading to a reduction in cellular survival, differentiation, and angiogenesis through the inhibition of GSK3β-mediated stabilization of β-catenin. Further inhibition of AKT activation leads to the inhibition of transcription factors for inflammatory genes through TSC1/2 and mTORC1 inhibition, thus contributing to the immunomodulatory, anti-inflammatory, and neuroprotective actions of curcumin [272].

Furthermore, curcumin has been shown to inhibit the adherence of *Streptococcus* mutants to human dental surfaces and the extracellular matrix protein [273]. Curcumin suppresses the expression of *Pseudomonas aeruginosa* (PAO1) virulence factors, including biofilm formation, pyocyanin biosynthesis, elastase/protease activity, and acyl homoserine lactone (HSL) production [274]. The antimicrobial action of curcumin is multifactorial, including the disruption of cell membrane function, the inhibition of bacterial cell division by binding to FtsZ, the induction of bacterial apoptosis, and the induction of phototoxicity due to a photosensitizer under blue-light irradiation [275]. As shown in Figure 4, curcumin reduces the bacterial virulence due to the instability of the biofilm structure, increased susceptibility to antibiotics, and the host immune system through the inhibition of a biofilm promotor gene [276]. In addition, it also prevents bacterial growth by disrupting the membrane structures and inhibits DNA gyrase, which in turn leads to bacterial lysis. Curcumin’s several antibacterial modes of action have made it a promising candidate for overcoming bacterial resistance and serving as an effective addition to antimicrobial therapy. However, the issues of its instability and low oral bioavailability from conventional dosage forms must be overcome in order to achieve the appropriate outcomes for therapeutic purposes.

## 6. Future Perspectives

While curcumin has demonstrated extensive therapeutic potential across various chronic diseases, several critical areas require further investigation to fully harness its benefits. Enhancing bioavailability remains a paramount challenge; innovative delivery systems, such as nanotechnology-based formulations, liposomal encapsulations, and curcumin–piperine combinations, show promise in overcoming curcumin’s poor absorption and rapid metabolism. Future studies should focus on optimizing these formulations to maximize bio-efficacy and ensure consistent therapeutic outcomes.

Large-scale clinical trials are essential to validate the promising preclinical findings and establish standardized dosing regimens. Most existing clinical studies on curcumin are limited by small sample sizes and short durations. Comprehensive, multi-center trials are necessary to assess curcumin’s long-term safety, efficacy, and role as an adjunct therapy in managing diseases, like cancer, diabetes, neurodegenerative disorders, and inflammatory conditions.

Combination therapies represent a promising avenue, where curcumin is used alongside conventional medications to enhance efficacy and reduce side effects. Investigating synergistic effects with existing drugs could lead to more effective treatment protocols, particularly in combating antibiotic resistance and enhancing antimalarial strategies.

Standardization and quality control of curcumin supplements are important to ensure consistency and reliability in clinical outcomes. Establishing standardized extraction methods and formulation protocols will enhance the reproducibility of research findings and facilitate regulatory approval.

Taken together, bridging the existing knowledge gaps through targeted research initiatives, innovative formulation strategies, and rigorous clinical evaluations will significantly advance curcumin’s role in modern medicine. These efforts will not only validate curcumin’s therapeutic efficacy but also pave the way for its integration into mainstream treatment protocols for a wide array of chronic diseases.

## 7. Conclusions

The health-promoting effects of curcumin are well recognized and have been in practice in traditional medicine since ancient times. Curcumin is used in traditional system-based medicinal products for diseases management because it is non-toxic and has only slight side effects. Numerous in vitro and in vivo laboratory based studies, as well as human clinical trials, have demonstrated the effectiveness of turmeric and its constituents as a favorable modulator of biological processes. Curcumin has shown a diverse range of effects in inflammatory disorders, perhaps by influencing multiple genes, signaling molecules, and enzymes. However, additional research is necessary to enlighten ourselves with the knowledge of curcumin’s wide-ranging benefits, efficiency, plausible uses, and working mechanisms, especially in terms of its usage in the prevention and treatment of human illnesses. The combination of curcumin with other potential therapeutic reagents should further be explored and tested in controlled clinical studies.

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
