# Peer review of "The Dynamic Role of Curcumin in Mitigating Human Illnesses: Recent Advances in Therapeutic Applications"

_pharmaceuticals, 2024, doi:10.3390/ph17121674_

Round 1
Reviewer 1 Report
Comments and Suggestions for Authors
The article submitted for review is very extensive, it has 40 printed pages, which corresponds to the volume of the monograph. The material is logically composed, but in order to comprehensively cover the topic of curcumin and analogues, the Authors should start from the beginning, i.e. by determining the structure of curcumin and confirming its structure through synthesis. Removing this deficiency will slightly only affect the existing structure of work. On page 2, around the 5th line from the bottom, it should be added: "The structure of curcumin, supported by its synthesis, was published as early as 1910 by Kostanecki and coworkers [reference number here]"
Miłobędzka, J.; von Kostanecki, S.; Lampe V. Zur Kentniss des Curkumins, Berichte der Deutschen Chemischen Gesselschaft, 43(2).1910, 2163-2170
Subsequent chapters of the paper document the effect of curcumin or turmeric on selected types of diseases along with a description of the alleged mechanisms of action.
The study also used a water-soluble modification: curcumin sulfate, which should be more correctly called curcumin monosulfuric acid or its salts with selected amines. This information should be specified on page 8 (17th line from the top and 15th line from the bottom).
I have a very small remark about the continuity of the inscription "Curkumin" in Figure 3.
The work is written clearly, I have no more critical comments on it, after minor corrections it should be published.
Author Response
The article submitted for review is very extensive, it has 40 printed pages, which corresponds to the volume of the monograph. The material is logically composed, but in order to comprehensively cover the topic of curcumin and analogues, the Authors should start from the beginning, i.e. by determining the structure of curcumin and confirming its structure through synthesis. Removing this deficiency will slightly only affect the existing structure of work. On page 2, around the 5th line from the bottom, it should be added: "The structure of curcumin, supported by its synthesis, was published as early as 1910 by Kostanecki and coworkers [reference number here]"
Miłobędzka, J.; von Kostanecki, S.; Lampe V. Zur Kentniss des Curkumins, Berichte der Deutschen Chemischen Gesselschaft, 43(2).1910, 2163-2170
Reply: Thank you very much for your insightful comments. We have modified the manuscript and added the synthetic method, as suggested.
Subsequent chapters of the paper document the effect of curcumin or turmeric on selected types of diseases along with a description of the alleged mechanisms of action.
Reply: The manuscript was revised as suggested.
The study also used a water-soluble modification: curcumin sulfate, which should be more correctly called curcumin monosulfuric acid or its salts with selected amines. This information should be specified on page 8 (17th line from the top and 15th line from the bottom).
Reply: We have updated the text.
I have a very small remark about the continuity of the inscription "Curkumin" in Figure 3.
Reply: It was rectified.
The work is written clearly, I have no more critical comments on it, after minor corrections it should be published.
Reply: Thank you very much for your valuable remarks and improvements suggestions.

Reviewer 2 Report
Comments and Suggestions for Authors
This manuscript introduced the molecular mechanisms of curcumin and its therapeutic applications in chronic health conditions. However, this manuscript was lack of a clear framework and the writing logic is not clear, just citing a large of preclinical or clinical trails of curcumin. Thus, no constructive guidance was provided in this manuscript and it may not meet the requirements of “Pharmaceuticals”. Some main comments are listed below:
1. In the part of “Preclinical and clinical evidences for therapeutic applications of curcumin”,the authors listed a number of preclinical and clinical trials, but the manuscript lacked central arguments in many sections of the diseases, and there was little output of the authors' views. Only listing the arguments is difficult to cause effective thinking, so the manuscript lacks reference value.
2. The title of this manuscript is “The Dynamic Role of Curcumin in Mitigating Human Illnesses: Recent Advances in Therapeutic Applications”,However, the diseases described in the part of “Preclinical and clinical evidences for therapeutic applications of curcumin” did not include all the diseases used by curcumin, such as diabetes and malaria. So it is suggested to rearrange the content of this part and make it correspond to the part of “Summary of Key Findings in Studies of Curcumin as a Therapeutic Agent”. In addition, curcumin treatment of certain diseases rely on the same mechanism, it is suggested that this section be classified according to treatment mechanism to make the structure clearer.
3. In the part of “Abstract”, it was written that “This article is focused on providing a better perspective into molecular mechanisms for possible actions along with an in-depth review of recent studies of curcumin, its beneficial role and therapeutic applications in chronic health conditions, with a focus on its cancer, inflammatory bowel disease, osteoarthritis, atherosclerosis, peptic ulcer, Covid19, psoriasis, vitiligo, and depression. ” Please state the reason for using the term “chronic health conditions” and explain whether diseases such as cancer described below fall under the category of chronic health conditions.
4. In the part of “Inflammatory bowel disease”, it was written that “As mentioned above, curcumin has anti-inflammatory qualities because of its ability to interact with toll-like receptors, a crucial step for innate immunity.” However, none of the examples listed in this section mentioned this mechanism. It is suggested to supplement a detailed mechanism of curcumin for the treatment of inflammatory bowel disease.
5. In the part of “Peptic ulcer”, it was written that “Antibiotics, histamine receptor blockers, and proton pump inhibitors are the preferred treatments for peptic ulcers.” However, this sentence is not related to the center of the full text, and it is recommended to delete.
6. In the part of “Colorectal cancer”, the description of the first clinical trial was too lengthy, and it is suggested to simplify the description of the study content to facilitate understanding.
7. In the part of “Probable Mechanisms of Action of Curcumin”, it was written that “Figure 2, below, illustrates action mechanisms of curcumin in diverse pathologic conditions.” However, the figure did not adequately show the mechanism described above, such as the antibacterial action of curcumin. It is suggested to supplement the content in the figure.
8. It is recommended to add the full definition when the abbreviation first appears, such as “ESR”, “CPR” and so on.
9. In the part of “Probable Mechanisms of Action of Curcumin”, it is suggested to check and correct the layout of the last paragraph.
Comments on the Quality of English LanguageMinor editing of English language required.
Author Response
This manuscript introduced the molecular mechanisms of curcumin and its therapeutic applications in chronic health conditions. However, this manuscript was lack of a clear framework and the writing logic is not clear, just citing a large of preclinical or clinical trails of curcumin. Thus, no constructive guidance was provided in this manuscript and it may not meet the requirements of “Pharmaceuticals”. Some main comments are listed below:
- In the part of “Preclinical and clinical evidences for therapeutic applications of curcumin”, the authors listed a number of preclinical and clinical trials, but the manuscript lacked central arguments in many sections of the diseases, and there was little output of the authors' views. Only listing the arguments is difficult to cause effective thinking, so the manuscript lacks reference value.
Reply: Thank you very much for your comments. We appreciate the thoughtful comments. The text has been revised to incorporate the authors' viewpoints on curcumin's potential for therapeutic use as well as a more critical evaluation and synthesis of the preclinical and clinical data. This will improve the manuscript's reference value and make the main points more understandable.
- The title of this manuscript is “The Dynamic Role of Curcumin in Mitigating Human Illnesses: Recent Advances in Therapeutic Applications”, However, the diseases described in the part of “Preclinical and clinical evidences for therapeutic applications of curcumin” did not include all the diseases used by curcumin, such as diabetes and malaria. So it is suggested to rearrange the content of this part and make it correspond to the part of “Summary of Key Findings in Studies of Curcumin as a Therapeutic Agent”. In addition, curcumin treatment of certain diseases rely on the same mechanism, it is suggested that this section be classified according to treatment mechanism to make the structure clearer.
Reply: Thank you for the thoughtful suggestion by the esteemed reviewer. The manuscript has been modified to include the diseases mentioned in the "Preclinical and Clinical Evidence for Therapeutic Applications of Curcumin" as suggested.
- In the part of “Abstract”, it was written that “This article is focused on providing a better perspective into molecular mechanisms for possible actions along with an in-depth review of recent studies of curcumin, its beneficial role and therapeutic applications in chronic health conditions, with a focus on its cancer, inflammatory bowel disease, osteoarthritis, atherosclerosis, peptic ulcer, Covid19, psoriasis, vitiligo, and depression. ” Please state the reason for using the term “chronic health conditions” and explain whether diseases such as cancer described below fall under the category of chronic health conditions.
Reply: The abstract has been modified to align the contents as per the objective of this review.
- In the part of “Inflammatory bowel disease”, it was written that “As mentioned above, curcumin has anti-inflammatory qualities because of its ability to interact with toll-like receptors, a crucial step for innate immunity.” However, none of the examples listed in this section mentioned this mechanism. It is suggested to supplement a detailed mechanism of curcumin for the treatment of inflammatory bowel disease.
Reply: Thank you for the valuable suggestion. The suggested mechanism of curcumin for the treatment of inflammatory conditions has been added in the manuscript under the mechanism section.
- In the part of “Peptic ulcer”, it was written that “Antibiotics, histamine receptor blockers, and proton pump inhibitors are the preferred treatments for peptic ulcers.” However, this sentence is not related to the center of the full text, and it is recommended to delete.
Reply: Modified as suggested.
- In the part of “Colorectal cancer”, the description of the first clinical trial was too lengthy, and it is suggested to simplify the description of the study content to facilitate understanding.
Reply: We deleted unnecessary text and refined the content as suggested.
- In the part of “Probable Mechanisms of Action of Curcumin”, it was written that “Figure 2, below, illustrates action mechanisms of curcumin in diverse pathologic conditions.” However, the figure did not adequately show the mechanism described above, such as the antibacterial action of curcumin. It is suggested to supplement the content in the figure.
Reply: Additional figures showing the mechanism of antibacterial action have been incorporated as suggested.
- It is recommended to add the full definition when the abbreviation first appears, such as “ESR”, “CPR” and so on.
Reply: Modified as suggested.
- In the part of “Probable Mechanisms of Action of Curcumin”, it is suggested to check and correct the layout of the last paragraph.
Reply: The content has been revised as suggested.

Reviewer 3 Report
Comments and Suggestions for Authors
Manuscript ID: pharmaceuticals-3243203 -The review article provides a detailed account of the historical and contemporary uses of Curcuma longa (turmeric) in traditional medicine, emphasizing both its cultural significance and therapeutic properties. The discussion of curcumin’s antioxidant and anti-inflammatory qualities is especially pertinent, given the current trend towards using natural compounds for managing chronic conditions.
Emerging Applications: The discussion of curcumin's role in addressing conditions like Covid-19, Alzheimer's, and Parkinson's is relevant and aligns with current interest in repurposing natural compounds. However, the review could be strengthened by integrating recent clinical trial data on curcumin's effectiveness in these areas, providing a more current perspective on its potential benefits also it would be better if authors can include curcumin nanoparticles bioavailability challenges as it is also an emerging area in the field of nanobiotechnology.
Considerations for Safety and Dosage: The article notes curcumin's general safety, even at high doses, but briefly mentions its anticoagulant properties. Expanding on the current research around curcumin’s pharmacokinetics, variable absorption rates among individuals, and interactions with other medications would offer a more nuanced view of its safe use in clinical applications.
Exploration of Combination Therapies: While the review alludes to combining curcumin with other therapeutic agents, it lacks detailed examples or references to recent studies. Further discussion on combination therapies could provide insights into potential synergistic effects, an area that has gained significant interest in recent research.
The review provides a good summary of curcumin’s therapeutic potential. It would be improved by incorporating more recent research trends, providing a deeper analysis of molecular mechanisms, and discussing advancements in curcumin formulations and combination therapies. These additions would offer a more comprehensive and updated perspective on the evolving field of curcumin research.
Comments on the Quality of English LanguageModerate English editing with a native English speaker is required. So, please go through the review with a native speaker
Author Response
The review article provides a detailed account of the historical and contemporary uses of Curcuma longa (turmeric) in traditional medicine, emphasizing both its cultural significance and therapeutic properties. The discussion of curcumin’s antioxidant and anti-inflammatory qualities is especially pertinent, given the current trend towards using natural compounds for managing chronic conditions.
Reply: Thank you very much for your comments. We appreciate the thoughtful comments.
Emerging Applications: The discussion of curcumin's role in addressing conditions like Covid-19, Alzheimer's, and Parkinson's is relevant and aligns with current interest in repurposing natural compounds. However, the review could be strengthened by integrating recent clinical trial data on curcumin's effectiveness in these areas, providing a more current perspective on its potential benefits also it would be better if authors can include curcumin nanoparticles bioavailability challenges as it is also an emerging area in the field of nanobiotechnology.
Reply: We appreciate your thoughtful suggestions. The contents of each section have been revised, and additional contents on Alzheimer's and Parkinson's have been incorporated under the neuroprotective section.
Considerations for Safety and Dosage: The article notes curcumin's general safety, even at high doses, but briefly mentions its anticoagulant properties. Expanding on the current research around curcumin’s pharmacokinetics, variable absorption rates among individuals, and interactions with other medications would offer a more nuanced view of its safe use in clinical applications.
Reply: Modified as suggested.
Exploration of Combination Therapies: While the review alludes to combining curcumin with other therapeutic agents, it lacks detailed examples or references to recent studies. Further discussion on combination therapies could provide insights into potential synergistic effects, an area that has gained significant interest in recent research.
Reply: Thank you for your valuable feedback and insightful comments. We appreciate your suggestion to include more detailed examples and references regarding combination therapies. In this study, we primarily focused on exploring the molecular mechanisms underlying the effects of curcumin, with limited discussion on combination therapy. However, we recognize the importance of combination therapies and agree that a more in-depth discussion of potential synergistic effects would add significant value, though it is beyond the scope of the current submission.
We would like to inform you that a more comprehensive review on combination therapies, including curcumin and other therapeutic agents, is currently in progress and will be submitted as a separate communication in the near future. This upcoming work will provide a detailed analysis of recent studies, highlight key findings, and further explore the potential of combination therapies in the context of our research.
The review provides a good summary of curcumin’s therapeutic potential. It would be improved by incorporating more recent research trends, providing a deeper analysis of molecular mechanisms, and discussing advancements in curcumin formulations and combination therapies. These additions would offer a more comprehensive and updated perspective on the evolving field of curcumin research.
Reply: Thank you very much for your insightful suggestion to improve the manuscript.

Round 2
Reviewer 2 Report
Comments and Suggestions for Authors
No.
Reviewer 3 Report
Comments and Suggestions for Authors
I feel the author answered all the revision queries raised in the last revision. So I recommend this manuscript for publication